# Molecular Epidemiology of Sapovirus in Children Living in the Northwest Amazon Region

**DOI:** 10.3390/pathogens10080965

**Published:** 2021-07-30

**Authors:** Marcia Terezinha Baroni de Moraes, Gabriel Azevedo Alves Leitão, Alberto Ignácio Olivares Olivares, Maria da Penha Trindade Pinheiro Xavier, Romanul de Souza Bispo, Sumit Sharma, José Paulo Gagliardi Leite, Lennart Svensson, Johan Nordgren

**Affiliations:** 1Laboratory of Comparative and Environmental Virology, Oswaldo Cruz Institute, Oswaldo Cruz Foundation, Rio de Janeiro 21040-360, RJ, Brazil; gabriel.leitao@ioc.fiocruz.br (G.A.A.L.); tpx@ioc.fiocruz.br (M.d.P.T.P.X.); jpgleite@ioc.fiocruz.br (J.P.G.L.); 2Secretaria Estadual de Saúde de Roraima, SESAU/RR, Boa Vista 69310-043, RR, Brazil; albertoufrr@gmail.com; 3Research Center Roraima Health Observatory (ObservaRR), Federal University of Roraima, Boa Vista 69310-000, RR, Brazil; romanulbispo@hotmail.com; 4Department of Biomedical and Clinical Sciences, Division of Molecular Medicine and Virology, Linköping University, 581 85 Linköping, Sweden; sumit.sharma@liu.se (S.S.); lennart.t.svensson@liu.se (L.S.); johan.nordgren@liu.se (J.N.); 5Department of Medicine, Division of Infectious Diseases, Karolinska Institute, 171 76 Stockholm, Sweden

**Keywords:** Sapovirus, histo-blood group antigen, Amazon region

## Abstract

Sapovirus is an important etiological agent of acute gastroenteritis (AGE), mainly in children under 5 years old living in lower-income communities. Eighteen identified sapovirus genotypes have been observed to infect humans. The aim of this study was to identify sapovirus genotypes circulating in the Amazon region. Twenty-eight samples were successfully genotyped using partial sequencing of the capsid gene. The genotypes identified were GI.1 (*n* = 3), GI.2 (*n* = 7), GII.1 (*n* = 1), GII.2 (*n* = 1), GII.3 (*n* = 5), GII.5 (*n* = 1), and GIV.1 (*n* = 10). The GIV genotype was the most detected genotype (35.7%, 10/28). The phylogenetic analysis identified sapovirus genotypes that had no similarity with other strains reported from Brazil, indicating that these genotypes may have entered the Amazon region via intense tourism in the Amazon rainforest. No association between histo-blood group antigen expression and sapovirus infection was observed.

## 1. Introduction

Sapoviruses belong to the genus *Sapovirus* within the family *Caliciviridae*, being responsible for both sporadic cases and occasional outbreaks of acute gastroenteritis (AGE). In lower-income communities, sapovirus frequencies in children under five years of age can be up to 17%, resulting in hospitalizations and severe dehydration [1].

The sapovirus genome comprises a positive-sense, single-stranded RNA genome, which is approximately 7.1 to 7.7 kb in size and contains two open reading frames (ORFs). ORF1 encodes a large polyprotein containing the nonstructural proteins (including the RNA-dependent RNA polymerase, RdRp) followed by the major capsid viral protein, VP1. ORF2 is predicted to encode the minor structural protein VP2 [2]. For the genetic classification of sapovirus, VP1 sequences are widely used because this region is more diverse than the RdRp region [3]. Based on complete VP1 nucleotide sequences, sapoviruses are classified into 19 genogroups, of which viruses from GI, GII, GIV, and GV infect humans and can be further subdivided into at least 18 genotypes [4]. Human sapovirus GI and GII genogroups are the most frequently detected worldwide in recent years [5,6,7]. The GIV genogroup is relatively rare, but it can at times be the third most common genogroup detected locally, as reported in Spain, Guatemala, South Africa, Canada [6,8,9,10], and recently infecting Indigenous infants in North America [11]. In South America, the detection of GIV.1 was reported only in Peru [12] and Venezuela [13]. GV is also rarely detected and was first detected in Argentina in 1995 [14].

The genotype diversity detected in low-income communities may play a significant role in the burden of AGE [15,16], and subclinical infections and diarrhea in children seem to give a minimal degree of protection to sapovirus [17]. Children less than 5 years old living in the low-income communities of the Amazon region are highly affected by AGE, and human sapoviruses have been observed with considerable diversity [18,19,20].

The histo-blood group antigens (HBGAs) affect the individual’s susceptibility to infection by norovirus, in a genotype specific manner [21]. Although sapoviruses belong to the same *Caliciviridae* family as noroviruses, and share the common characteristics of the viral family, the few studies that have investigated this have reported no association between HBGA and human sapovirus susceptibility [22,23,24].

The aim of this study was to identify the sapovirus genotypes circulating in younger children ≤ 5 years old living in the Amazon region, including those in isolated areas from the Amazon rainforest (Brazil and Venezuela), detected during an epidemiological investigation study of viruses causing AGE in 2016–2017 [25] The detected sapovirus genotypes were associated with clinical and epidemiological parameters as well as HBGA.

## 2. Results

### 2.1. Genetically Diverse Sapoviruses Cause Acute Gastroenteritis in Children in the Amazon Region

Mixed infections with rotavirus A (RVA), norovirus, human adenovirus (HAdV), and human bocavirus (HBoV) were detected in some of the 52 sapovirus-positive samples as previously reported (25). Children infected exclusively with sapovirus (*n* = 12) presenting AGE were 1.8% (9/485) of the total and those of the control group were 1.2% (3/249). Figure 1 represents the occurrence of sapovirus infection in children living in the Amazon region according to age.

Children between 6 and 12 months of age were largely infected by sapovirus. No sapovirus infection alone was detected in children ≤3 months old and between 2 and 5 years of age. Children infected exclusively with sapovirus were mostly Rotarix (RV1) vaccinated with two doses (data not shown). Seventy percent of the sapovirus-positive samples (28/40) were successfully genotyped using one of the PCR amplification strategies described in Table 1.

The initial strategy using the primer pair SLV5317-F and SLV5749-F and one round of amplification generated amplicons suitable for Sanger sequencing of more than half of samples (57%, 16/28). Three samples were amplified only by the primer pair HuSaV-5159-F and HuSaV5498-R (strategy sixty, one round). The GI, GII, and GIV sapovirus genogroups that were detected comprised GI.1, GI.2, GII.1, GII.2, GII.3, GII.5, and GIV.1 genotypes (Figure 2). Table 2 shows some clinic-epidemiological features concerning each genotype.

The most prevalent genotypes, GIV.1 and GI.2, were distributed across the northern part of the map; however, GIV.1 was detected only in the Brazilian Amazon region. The GII.3 genotype was mostly detected in Boa Vista (urban area) in RR state.

### 2.2. Phylogenetic Analysis Shows the Heterogeneity of Sapoviruses in the Amazon Region

Nucleotide sequences retrieved from GenBank were selected based on the highest similarity to the Amazon nucleotide sequences of the strains sequenced in the present study. The phylogenetic tree in Figure 3 shows the heterogeneity of sapovirus genotypes circulating in the Amazon region.

The GIV.1 Amazon region sequences clustered together with the reference samples from Venezuela. The GI.2 genotype detected in the Amazon region was clustered separately from the reference nucleotide sequences from Canada (KU973908.1 and KU973906.1) and USA (MN486490.1). Similarly, the GII.3 genotype was clustered separately from the reference sequence from Peru (MG012417.1). The GI.1 and GII.1 genotypes detected in the Amazon region were very similar to the reference sequences from Thailand (AY646853.2) and Russia (MF589697.1). The quality of the GII.5 nucleotide sequence from the Amazon region was not suitable for phylogenetic analysis.

### 2.3. No Association between Sapovirus Genotypes and Histo-Blood Group Susceptibility

The HBGA profile regarding the secretor/Lewis status for all 28 genotyped samples is shown in Table 2. The HBGA data were previously obtained [25]. The HBGA results were heterogeneous, and no clear association between sapovirus genotypes and specific HBGAs was observed.

## 3. Discussion

Sapoviruses are now recognized as an important cause of AGE in children [3,29,30,31,32]. In this study, we investigated the genetic variability of sapoviruses in children in the Amazon region.

Genetic characterization of sapoviruses using molecular methods can be difficult due to the genetic variability of the region encoding the major structural VP1 protein, requiring the design of new primers. The application of primer-independent metagenomic sequencing approaches for identification of human sapovirus has also been reported recently and is able to characterize the presence of sapoviruses of different genogroups in feces from children with AGE [31]. We observed a heterogeneity and spread of sapovirus genotypes throughout the Amazon region where the children live. The prevalence was higher in the AGE group compared to the control group in children infected exclusively with sapovirus, but this was not significant (OR = 2.6 with a 95% CI of 0.6 to 11.9, *p* = 0.22).

To characterize the genotypes that circulated in the Amazon region, we used several pairs of primers in a single or two rounds of amplification of part of the region encoding the major structural VP1 protein. This VP1 region is the most variable and is used for genetic typing [33]. Additionally, one new primer pair was designed specifically considering the diversity of sapovirus samples as described previously [28]. Using these multiple sets of primers, three genogroups distributed in seven genotypes (GI.1, GI.2, GII.1, GII.2, GII.3, GII.5, and GIV.1) could be detected in this study. Seventy percent of the sapoviruses detected (28/40) were successfully genotyped. No correlation between sapovirus viral load (*C*_t_) and ability to generate a PCR amplicon for sequencing was observed (data not shown), suggesting a high genetic heterogenicity that was not detected, even using multiple sets of primers. Interestingly, the GIV.1 genotype was the most detected followed by GI.2 and GII.3. The GIV.1 genotype was detected in children in different municipalities of the states of RR and Amazonas. No GIV.1 sample was detected from the children living in Venezuela despite the GIV.1 genotype already having been detected in Venezuela [13]. Complete genome sequencing of samples genotyped as GIV.1 might elucidate the origin of GIV.1. The GIV.1 further predominately infected RV1 unvaccinated children with AGE and was the only sapovirus genotype detected in children with dehydration (2 children), indicating that may have an important role in AGE caused by sapovirus. Unfortunately, the number of sapovirus samples genotyped in Brazil are few. The genotypes here identified, except for GIV.1, have been detected in children under 5 years old with AGE living in different regions in Brazil, including the Amazon region [19,20,34]. The phylogenetic analysis showed greater similarity with sapoviruses from countries other than Brazil, which could be explained by the Amazon region having an intense influx of tourism from other countries.

The histo-blood group antigens, particularly secretor status, is important for susceptibility to norovirus, another member of the *Caliciviridae* family [21]. Previous observational studies in Burkina Faso and Nicaragua [22] [23], however, have not observed any such association with regards to sapovirus. In this study, accordingly, no association between secretor or Lewis status to sapovirus infection was observed.

## 4. Materials and Methods

### 4.1. Information Regarding the Samples of This Study

In an epidemiological investigation study to identify viral etiologic agents causing AGE and virus–host susceptibility in children living in the Amazon rainforest, 1468 samples were collected (feces and saliva collected in parallel) from 734 children ≤5 years old across the span of 1 year (October 2016 to October 2017) [25]. Fifty-two samples were sapovirus-positive, and forty archived samples were submitted to genotyping (a cycle threshold [*C***_t_**] under 37) for this study. These samples were collected from children presenting AGE (77.5%, 31/40) living in the Amazon region as well as a control group of children with respiratory symptoms (22.5%, 9/40). The ethnic indigenous groups of these children were described previously [25,29]. The collection site of the samples was the emergency care unit at the “Hospital da Criança de Santo Antonio” (HCSA) located in Boa Vista, state of Roraima (RR, Brazil). The HCSA is the only hospital placed in RR that attends children living in the extreme north of Brazil and borders Venezuela and Guyana, including those living in the Amazon rainforest in demarcated indigenous areas.

Most of the children enrolled in this study lived in the Brazilian and International Amazon rainforest, together with their parents, in indigenous areas including those in the demarcated area. AGE (cases) was clinically characterized according to the WHO criterion as: _AGE: ≥three liquid/semi-liquid evacuations in a 24 h period and dehydration; Control with other clinical symptoms not clinically characterized as AGE.

Each child was examined by a pediatrician, and the child’s parents or guardians were interviewed to collect data and fill out a form containing clinical and epidemiological information for each child including the RV1 vaccination information. All saliva samples were collected at least 1 h before or after breastfeeding. The fecal samples were previously tested by quantitative molecular detection for rotavirus A (RVA), norovirus genogroups GI and GII, human bocavirus (HBoV), human adenovirus (HAdV), and sapovirus [25].

### 4.2. Sapovirus Genotyping

RNA of forty archived sapovirus-positive samples [25] were subjected to cDNA synthesis using illustra™ Ready-to-go RT-PCR beads (GE HealthCare, Uppland, Uppsala, Sweden), according to the manufacturer’s instructions, with a random primer (2.5 µg) and 28 µL of total RNA extracted as previously described [25] using the QIAamp Viral RNA Mini Kit and QIAcube automated system (QIAGEN, Germantown, MD, USA). For single or multiplex PCR reactions, 10 µM of each set of previously described primers [26,27,28], or one new set primer designed for this study (Table 1), for the VP1 (ORF1) sapovirus region, was added to 2.5 µL of the cDNA in a tube containing a bead with DNA polymerase, M-MuLV reverse transcriptse, RNAse inhibitor, reaction buffer, stabilizers, and 200 µM of each dNTP and 1.5 mM MgCl_2_ (Illustra ™ PCR beads, GE HealthCare, Uppland, Uppsala, Sweden), for a final volume of 25 µL, according to the manufacturer’s instructions. A single or two amplification rounds were used with the following cycling program: initial denaturation at 94 °C for 5 min, followed by 40 or 50 cycles of 94 °C for 30 s, 50 °C or 53 °C for 30 s, 72 °C for 1 min, and a final extension at 72 °C for 7 min (Table 1). For the Sanger sequencing, the amplicons were sent to Macrogen Europe B.V. Company (North Holland, Amsterdam, The Netherlands) together with the same PCR primers used above as described for PCR amplicons (Table 1).

### 4.3. Histo-Blood Group Antigen Phenotyping

The saliva secretor and Lewis phenotypes had been characterized in a previous study [25], using enzyme immunoassay (EIA) for the detection of Lea, Leb, and Fucα1-2Gal-R as described [25,29]. In the present study, such results were considered to verify the association with sapoviruses genotyped here.

### 4.4. Phylogenetic Tree and Statistical Analysis

The chromatograms of the VP1 sapovirus coding nucleotide sequences were analyzed using the free tracer viewer Chromas 2.4 (Technelysium Pty Ltd., South Brisbane, Qheensland, Australia). VP1 nucleotide region and amino acid multiple alignment sequencing was done using the Mega Molecular Evolutionary Genetic Analysis Version X software and compared with reference nucleotide sequences of VP1 sapovirus available in the GenBank database of the National Center for Biotechnology Information (NCBI). Sequences were analyzed using the maximum likelihood method (Tamura–Nei model). The representative gene sequences of VP1 sapovirus obtained in the current study were submitted to GenBank under the access numbers MW349916 to MW349937 as shown in the phylogenetic tree (Figure 1). Statistica 12.6 software (December 2014; TIBCO Software Inc., Palo Alto, CA, USA) was used for all statistical analyses. The statistical tests, when appropriate, were Pearson chi-square or Fisher exact test and OR [30].

## 5. Conclusions

In conclusion, this study shows a high genetic diversity of sapoviruses that could be partly attributed to the intense tourism involving the Amazon rainforest. Sapovirus genotyping was improved using different sets of primers, which can be used for samples with low C_**t**_, increasing the genotyping success of these genetically diverse viruses. GIV.1, which has previously not been observed in Brazil, was here the most prevalent genotype and could be an emerging, important agent in causing AGE.

## Figures and Tables

**Figure 1 pathogens-10-00965-f001:**
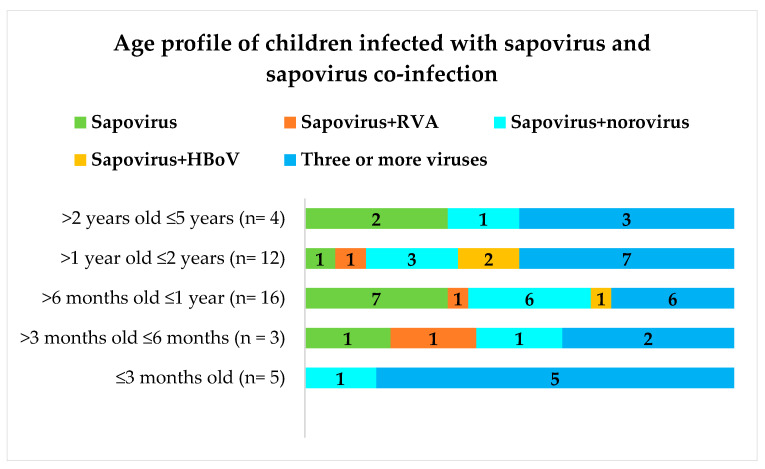
Age profiles of 52 sapovirus-positive children with and without co-infection. “Three or more viruses” corresponding to sapovirus co-infection with two from one of the following viruses: rotavirus A, norovirus, human bocavirus, and human adenovirus. Numbers within each bar of the graph correspond to the “*n*” of children.

**Figure 2 pathogens-10-00965-f002:**
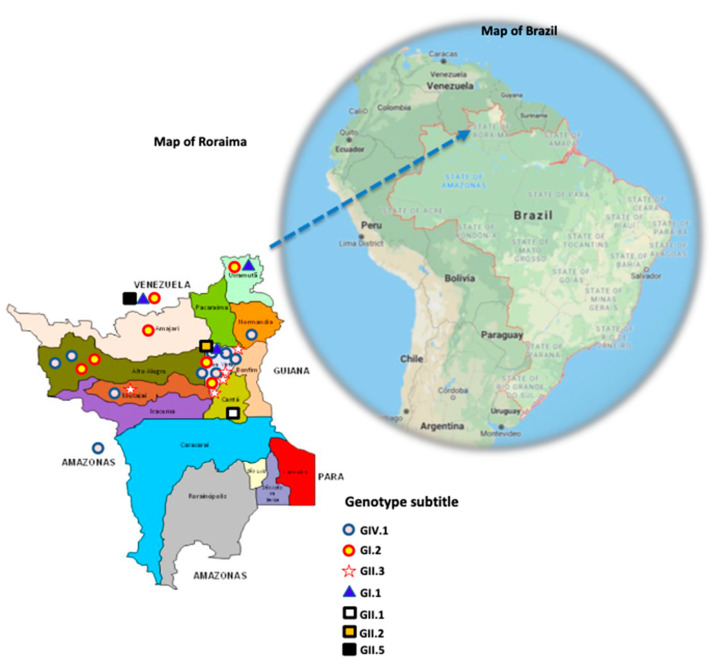
Map of Roraima state (the state of Amazonas and countries of Venezuela and Guyana are indicated outside this map), where the sapovirus genotypes are indicated according to the subtitles. Behind the map of Roraima, the map of Brazil is shown, where the state of Roraima is indicated by a blue dotted arrow (https://www.google.com/maps/place/Brazil/; accessed on 26 July 2021).

**Figure 3 pathogens-10-00965-f003:**
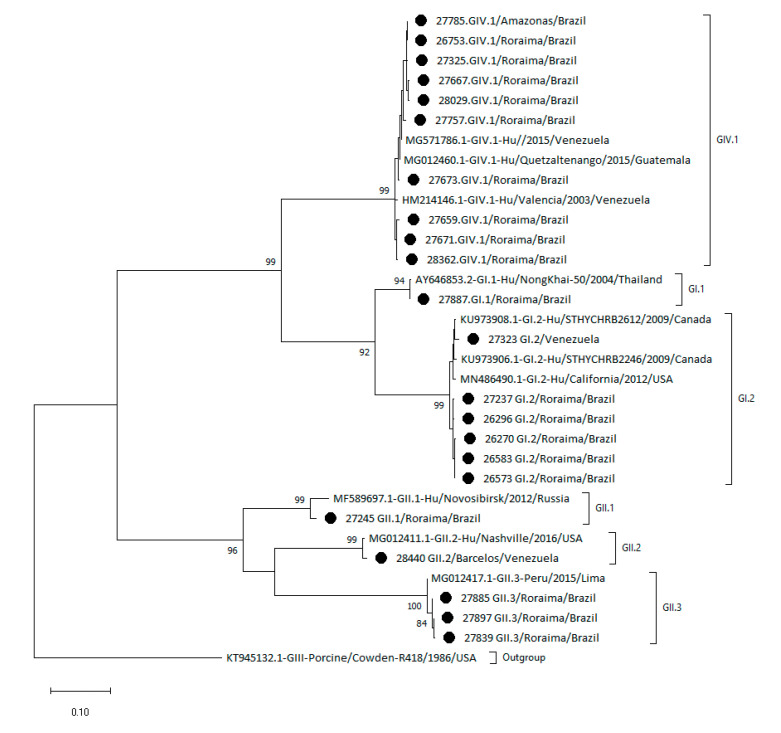
Phylogenetic tree of sapovirus based on partial capsid VP1 gene sequences from children from the Amazon region. Sequences were analyzed using the maximum likelihood method (Tamura–Nei model), and bootstrap values >80% are shown at the nodes of the tree as percentages on 2000 replicates. The strains reported in this study are indicated by circles. The reference strains are shown with their respective GenBank access numbers and each genotype outside of brackets.

**Table 1 pathogens-10-00965-t001:** Primers and PCR conditions used in this study to obtain sapovirus PCR amplicons.

PCR Amplification Strategy	Primer (Forward-F, Reverse-R, Round-1 ^1^, Round-2 ^1^, Multiplex)	Primer Sequence 5′–3′; (VP1 Nucleotide Position)	Annealing Temperature/Number of Cycles	Round Number	Reference
Initial	SLV5317-F	CTCGCCACCTACRAWGCBTGGTT	50 °C/40	1	[26]
	SLV5749-R	CGGRCYTCAAAVSTACCBCCCCA			
Second	1 ^1^ SV-F13-F	GAYYWGGCYCTCGCYACCTAC		2	[27]
	1 ^1^ SV-R13-R	GGTGANAYNCCATTKTCCAT			
	2 ^1^ SV-F22-F	SMWAWTAGTGTTTGARATG			
	2 ^1^ SV-R2-R	GWGGGRTCAACMCCWGGTGG			
Third	1 ^1^ SV-F14-F	GAACAAGCTGTGGCATGCTAC		2	[27]
	1 ^1^ SV-R14-R	GGTGAGMMYCCATTCTCCAT			
	2 ^1^ SV-F22-F	SMWAWTAGTGTTTGARATG			
	2 ^1^ SV-R2-R	GWGGGRTCAACMCCWGGTGG			
Fourth	Multiplex-SV-F13-F	---		1	[27]
	SV-F14-F	---			
	SV-G1-R-R	CCCBGGTGGKAYGACAGAAG			
	SV-G2-R-R	CCANCCAGCAAACATNGCRCT			
	SV-G4-R-R	GCGTAGCAGATCCCAGATAA			
	SV-G5-R-R	TTGGAGGWTGTTGCTCCTGTG			
Fifth	**SaVPanF-F**	**CAGTTCWACTGGSTNAAGGC** **(5047-5066)**		**1**	**This study**
	**SaVPanR-R**	**GCATCAACRAANGCGTGNGG (5816-5835)**			
Sixth	HuSaV-5159F-F	TAGTGTTTGARATGGARGG	53 °C/50	1	[28]
	HuSaV-5498R-R	CCCCANCCNGCVHACAT			
Seventh	Multiplex-SaV 1245Rfwd-F	TAGTGTTTGARATGGAGGG			
	SV-G1-R-R	---			
	SV-G2-R-R	---			
	SV-G4-R-R	---			
	SV-G5-R-R	---			
Eighth	HuSaV-F1-F	GGCHCTYGCCACCTAYAA YG			
	HuSaV-5498R-R	---			

^1^ The strategy number corresponds to the order in which it was used.

**Table 2 pathogens-10-00965-t002:** Sapovirus genotypes detected in children living in the Amazon region and some clinical and epidemiological features.

Genotype (*n*)	Child Age ^1^ (*n*)	Clinical Aspects ^2^	HBGA Status ^3^
GIV.1 (10)	>3 months ≤6 months (3)	Mucus in feces, fever, abdominal pain, dehydration	Lea+Leb−, *Se*−; Lea+Leb+, *Se*+
	>6 months ≤1 year (2)	Mucus in feces, fever, abdominal pain; mucus in feces, fever, vomit	Lea−Leb−, *Se*−; Lea−Leb+, *Se*+
	>1 year ≤2 years (3)	Mucus in feces, fever, vomit, abdominal pain	Lea−Leb+, *Se*+; Lea−Leb−, *Se*+; Lea−Leb−, *Se*−
	>2 years ≤5 years (2)	Mucus and blood in feces, vomit, abdominal pain; fever, abdominal pain	Lea+Leb+, *Se*+; Lea−Leb+, *Se*+
GI.2 (7)	>6 months ≤1 year (3)	Coryza, couch; mucus in feces, vomit, abdominal pain; mucus in feces, fever, vomit, abdominal pain	Lea+Leb+, *Se*+; Lea−Leb+, *Se*+
	>1 year ≤2 years (4)	Mucus in feces, fever, vomit, abdominal pain; mucus in feces, fever, abdominal pain; fever, couch	Lea−Leb+, *Se*+; Lea−Leb−, *Se*+
GII.3 (5)	≤3 months (1)	Fever, abdominal pain	Lea+Leb+, *Se*+
	>6 months ≤1 year (3)	Fever, vomit, cough; fever, abdominal pain, cough; mucus in feces, fever, abdominal pain	Lea+Leb+, *Se*+
	>2 years ≤5 years (1)	Mucus in feces, fever, vomit, abdominal pain	Lea−Leb+, *Se*+
GI.1 (3)	>6 months ≤1 year (2)	Fever, abdominal pain, cough; mucus in feces, fever, abdominal pain	Lea+Leb−, *Se*−; Lea+Leb−, *Se*+
	>2 years ≤5 years (1)	Mucus and blood in feces, fever, vomit, abdominal pain	Lea+Leb−, *Se*−
GII.1 (1)	>6 months ≤1 year	Mucus in feces, fever, abdominal pain	Lea+Leb+, *Se*+
GII.2 (1)	>1 year ≤2 years	Mucus in feces, fever, vomit, abdominal pain	Lea+Leb+, *Se*+
GII.5 (1)	≤3 months	Not available	Lea−Leb+, *Se*+

^1^ The groups of children enrolled in this study are a control group and a group with acute gastroenteritis; ^2^ Fever =≥ 38.5 °C; ^3^ HBGA = Histo-blood group antigens = Lewis and secretor status.

## Data Availability

The representative gene sequences of VP1 sapovirus obtained in the current study were submitted to GenBank under the access numbers MW349916 to MW349937.

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
