# Peer review of "Molecular Epidemiology of Sapovirus in Children Living in the Northwest Amazon Region"

_pathogens, 2021, doi:10.3390/pathogens10080965_

Round 1

Reviewer 1 Report

Marcia et al. proposed to identify SaV genotypes circulating in the Amazon region in the paper titled "Molecular epidemiology of sapovirus in children living in the Northwest Amazon region".

Although it currently present a serious flaw in terms of protocols not being stated clearly, the paper is of good quality and it will surely bring a real contribution to the current literature.

The abstract is clear and concise.

The introduction is well written and citations are used properly and sufficient.

Materials and methods should follow the Introduction and not be the last chapter. This chapter left for last is my only concern as it is insufficiently explained. The purpose of an article paper, beside it's novelty and contribution to the literature, is to be reproducible. 

4.2. Sapovirus genotyping 

Line 212-224: "RNA of Forty SaV positive samples [25] were subjected to cDNA synthesis using the illustra™ Ready-to-go RT-PCR beads (GE HealthCare, Uppsala Sweden), according to the manufacturer’s instructions with a random primer and 28µL of total RNA extracted as previously described [25].  "

Please properly state the protocol instead of vaguely cite it, also please delete the extra space after the dot.

4.3. Histo Blood Group antigen phenotyping 
Line 226-228: "The saliva secretor and Lewis phenotypes had been characterized in a previous study [25], using Enzyme Immunoassay (EIA) for the detection of Lea, Leb and Fucα1-2Gal-R 227 as described [25]."

The same issue, state the protocol used. The citation used in both cases ([25]) leads to a pre-proof version (https://www.sciencedirect.com/science/article/pii/S120197122100463X)

Are there any reasons as in why the protocols used in this work can not be clearly stated in an step by step manner?

If this crucial fault regarding vague protocol statement is fixed, as well as formatting and proper order of chapters accordingly to the journal style.

Reviewer 2 Report

This is a well-written paper on the circulation of sapovirus genotypes in young children in the Amazon region of Brazil and Venezuela and confirms previous studies that these viruses can also be detected in asymptomatic children, although the single sapovirus infections are low (1-2% range). The authors typed stool samples that had been collected from children presenting AGE living in the Amazon region, as well as a control group, of children with respiratory symptoms using existing and newly developed primer pairs. It is not completely clear if this study adds only genotyping to the already for sapovirus tested samples in a previous study. Please make that clearer in the materials and methods perhaps using words such as ‘archived’ stool samples. Currently, it somehow reads that these samples were collected for the current study.

The authors designed new sapovirus typing primers and provide an overall conclusion on the typing success, but it would be nice for those in the field that are interested in increasing the typing success, in particular for samples with low ct values, of these genetically diverse viruses to mention more information in the Conclusion section of the paper

It is not clear why the histoblood group typing was done and if this was done also in the previous study.

Sapoviruses are ubiquitous and can be detected worldwide so mentioning that GIV.1 was detected for the first time and comparing with other GIVs in the region should be shortened, and giving ‘primacy’ by saying ‘for the first time’ puts the burden on the reader to confirm this so would avoid.  Also, the statement that GIV primarily infected ‘unvaccinated’ children should be clarified by which vaccine the authors are referring to.

Overall, please proofread the manuscript as there a few minor issues such as not a full stop after sentences etc. Finally, single words such as sapovirus should not be abbreviated in the English language so it is ‘sapovirus’, not ‘SaV’.
